# Dual self-assembly of supramolecular peptide nanotubes to provide stabilisation in water

Julia Y. Rho [1], Henry Cox [2], Edward D.H. Mansfield[1], Sean H. Ellacott[1], Raoul Peltier[1], Johannes C. Brendel[1], Matthias Hartlieb[1], Thomas A. Waigh[2,3] & Sébastien Perrier [1,4,5]*

Self-assembling peptides have the ability to spontaneously aggregate into large ordered structures. The reversibility of the peptide hydrogen bonded supramolecular assembly make them tunable to a host of different applications, although it leaves them highly dynamic and prone to disassembly at the low concentration needed for biological applications. Here we demonstrate that a secondary hydrophobic interaction, near the peptide core, can stabilise the highly dynamic peptide bonds, without losing the vital solubility of the systems in aqueous conditions. This hierarchical self-assembly process can be used to stabilise a range of different β-sheet hydrogen bonded architectures.

[1] Department of Chemistry, University of Warwick, Coventry CV4 7AL, UK. [2] Biological Physics, School of Physics and Astronomy, University of Manchester, Manchester M13 9PL, UK. [3] Photon Science Institute, University of Manchester, Manchester M13 9PL, UK. [4] Faculty of Pharmacy and Pharmaceutical Sciences, Monash University, Parkville, VIC 3052, Australia. [5] Warwick Medical School, University of Warwick, Coventry CV4 7AL, UK. *email: S.perrier@warwick.ac.uk

Proteins play a vital role in all living systems, from the catalysis of metabolic reactions to providing structure of our cells. Since the hierarchical self-assembly of proteins was elucidated, these fascinating molecules have been the subject of a host of biological applications. The amino acid building blocks of the peptide (primary structure) determine how the sequence will non-covalently fold and interact with itself (secondary and tertiary structure) and other peptides (quaternary structure). Nucleases, tubular proteins such as the tobacco mosaic virus and the collagen of our connective tissues all use this hierarchical assembly to form their impressive complex and defined nanostructures. Although synthetically reproducing the complexity of these systems is still a far prospect, the advent of supramolecular polymers has brought us a step closer by providing us with a powerful tool to synthesise a range of different morphologies on the nanoscale, such as nanofibers[1], nanoribbons[2] and nanotubes[3,4]. These nanostructures consist of repeating subunits (unimers) that can self-assemble over long ranges via directional non-covalent interactions, most commonly π–π stacking[5], hydrogen bonding[6] and hydrophobic interactions[7]. Many of these systems have been optimised to form very large elongated assemblies, typically nanofibers, which create supramolecular gel networks[3,8–11]. Recent advancements in supramolecular systems have led to them being used in a wide range of applications, in optoelectronics[12,13], self-healable materials[14] or bio-therapeutics[15–17].

As a motif to form supramolecular polymers, hydrogen bonded core-stacking moiety have gained much attention due to their ability to form directional nanostructures[18–21]. The ease in functionalising these cores with a range of different substituents, such as polymers[22], dyes[23] or drugs[16], make them ideal candidates for bioactive materials such as antimicrobials[24] or drug delivery vectors[15]. The most prominent example of hydrogen bonded self-assemblies are peptides. The amino acid building blocks of peptides provide an extensively library of functional groups whilst the amide backbone can prove an in-built self-assembly motif, which can take part in hydrogen bonding. The simplest example is the dipeptide system[9,25]. Most notably, they have been shown to self-assembly to form networks of nanofibers, which have shown promising applications in tissue engineering[26] and drug delivery[27]. Another notable example is the surfactant based peptide amphiphiles which contains a hydrophilic tail, β-sheet forming middle segment and an outer bio-active epitope[6,28,29]. This two-fold self-assembly system has been used to develop bio-mimetic self-assembled nanofibers in order to aid bone mineralisation[30] and would healing[29].

Among the systems reported to date, an exciting supramolecular candidate for potential biological applications is the family of self-assembling cyclic peptide nanotubes (CPNT). Alternating D-octapeptide and L-octapeptide, upon cyclisation, have been shown to stack via hydrogen bonding to form a nanotubular assembly[4,31]. Due to the cyclic construction of the peptide this system also has more opportunity for functionalisation, compared to its linear counterparts. Previously, the multiple modification on the periphery of the cyclic peptide, either with different orthogonal polymers[7,32] or multiple polymer arms (1–3)[22] has been reported. Moreover, Xu and co-workers have shown that the interior of the cyclic peptide nanotubes can be tuned without compromising the ability to form nanotubular structures[33]. The attachment of hydrophilic polymers to the periphery of the peptide core has been studied to provide water soluble aggreates[22,34,35]. More recently, they have been shown to be pH and redox responsive[36], have antimicrobial activity[24,37] and be a promising drug carrier[15,16].

The transfer of supramolecular materials into applications requires a good understanding of their dynamic behaviour to avoid unexpected de-polymerisation or aggregation. Thus, the exchange behaviour of CPNT was studied using dye conjugated peptides in water and in vitro[23]. The peptide unimers were shown to rapidly exchange between the self-assembled nanotubes. The highly dynamic nature of these supramolecular assemblies explains why many of these systems are very short in length, typically around 10 nm[16,38]. The hydrophilic polymers arms decorating the peptide nanotube provide a steric barrier to prevent lateral aggregation and in turn improve their solubility in water, however dramatically lower their aggregation number[22,36].

The next generation of bio-inspired nanomaterials aims to encompass multiple design features, including controlled size, stability, bio-compatibility and high functionality. There are only a few examples of stabilised elongated uniform self-assembled nanostructures using the concept of living supramolecular polymerisation. One approach established and developed by Manners, Winnik and co-workers uses crystallisation-driven self-assembly (CDSA). By using a crystallisable core, such as poly-ferrocenylsilane (PFS), one can form kinetically trapped assemblies which can be chain extended with free unimers to produce uniform supramolecular block copolymers[39–41]. The second system, established by Sugiyasu, Takeushi and co-workers uses zinc or copper complexed porphyrin molecules to form seeds which upon nucleation polymerises further with additional unimers. This pathway controlled mechanism is able to produce uniform fibres in a precise manner[42]. The third system by Aida and co-workers consists of metastable cage-like monomer species containing hydrogen bond donor sites, which, upon heating could be opened by a hydrogen bond acceptor containing initiator[43]. As the monomer changes its geometry, the resulting initiator-monomer complex reveals the new hydrogen bond acceptor sites. This hydrogen bond reorganisation leads to a growing chain end that can propagate in a living fashion. These approaches illustrate controlled supramolecular polymerisation using highly specialised unimers and strict conditions. Despite these advances, designing controlled self-assembly systems for biological relevant systems in aqueous conditions has been much more challenging.

Inspired by the hierarchical design and structure of proteins, a two-fold self-assembly approach was developed to help stabilise the elongated peptide nanostructure. The hydrogen bond stacking of the cyclic peptides provides the primary structure and overall cylindrical morphology of the self-assembled aggregates. Here, we introduce a secondary structural driving force in the form of a hydrophobic region around the peptide to stabilise the single cyclic peptide nanotubes. The individual nanotubes remain independent due to the hydrophilic corona stabilising the nanostructures in water. This hierarchical approach offers a method to stabilise, not only cyclic peptides, but other β-sheet forming self-assembled architectures. The increasing complexity of these systems is the next step to realising the biological applications of these promising synthetic supramolecular polymers.

## Results

**Conjugates synthesis.** In order to stabilise and improve the self-assembly of cyclic peptide-polymer nanotubes, an amphiphilic diblock co-polymer was attached to the periphery of the CP. As sterically demanding polymers have been shown to dramatically destabilise the nanotubes[22] less bulky monomers such as butyl acrylate (BA) and dimethyl acrylamide (DMA) were chosen to form the hydrophobic and hydrophilic blocks, respectively. In order to form discrete regions around the cyclic peptide core (Fig. 1), reversible addition fragmentation chain-transfer (RAFT) polymerisation was employed to generate narrow dispersity ($Đ$ < 1.15) diblock co-polymers (pBA-*b*-pDMA, **1**)[44,45]. A homopolymer of the hydrophilic monomer of the same length (i.e. degree of polymerisation) lacking the hydrophobic region was also synthesised as a control (pDMA, **2**).

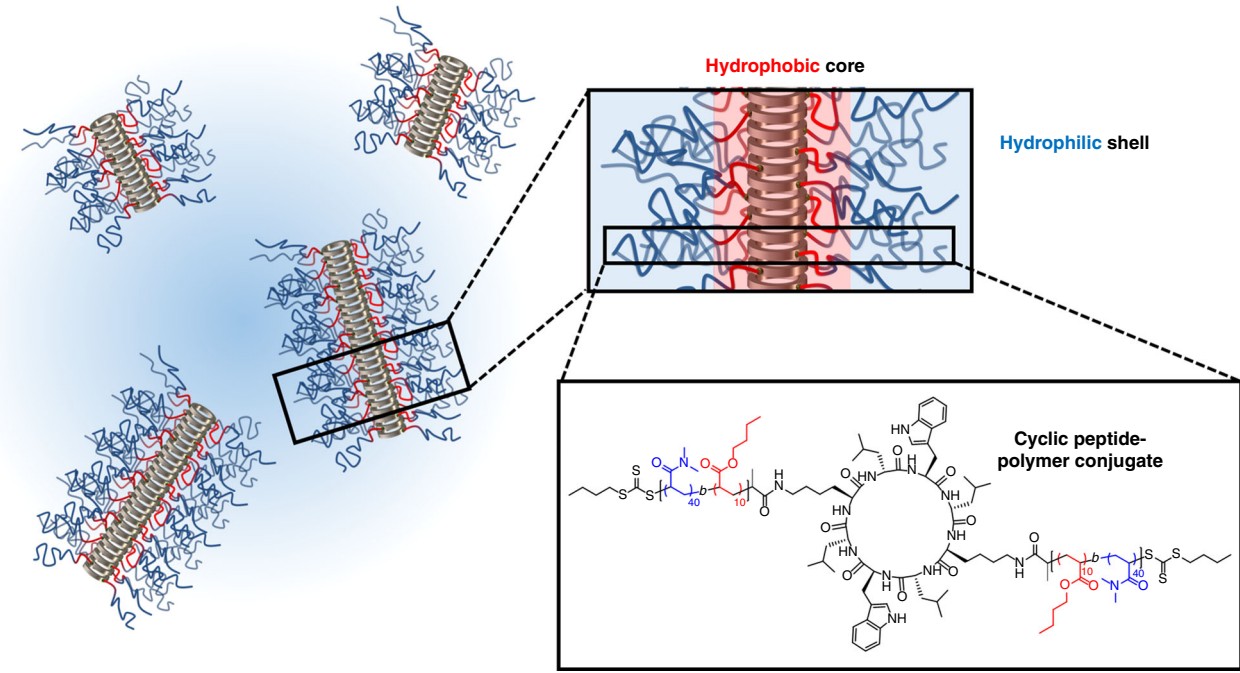

**Fig. 1** Self-assembling cyclic peptide-diblock polymer conjugates. The conjugation of carboxylic acid diblock co-polymer (pBA-*b*-pDMA, **1**) with the diamine functionalised cyclic peptide. In aqueous conditions, the conjugated cyclic peptide-diblock conjugate (CP-(pBA-pDMA)$_2$, **3**) self-assembles to form a hydrophobic core around the peptide and a hydrophilic corona

A CP with two conjugation sites was chosen to provide a dense hydrophobic core region around the peptide to better exclude water molecules competing with the hydrogen bond stacking of the nanotubes. The amine units of lysine were used to provide the handles for post-reaction modification of the peptide with the desired polymers, see Supplementary Fig. 19. In the size exclusion chromatography (SEC) trace, a clear double molecular weight shift between the free polymer and the two-arm cyclic peptide-polymer conjugate was observed (see Supplementary Figs. 16 and 18). In the reaction mixture, containing both the conjugate and excess polymer, two distributions are visible. After purification via precipitation the lower molecular weight distribution associated with unconjugated polymer is removed. A conjugate with a similar degree of polymerisation, but without a hydrophobic region, was also synthesised as a control. The detailed synthetic procedure is given in Supplementary Fig. 4.

**Self-assembly characterisation**. Next, the morphology, dimensions, toxicity and self-assembly mechanism of these self-assembling cyclic peptide-polymer nanotubes were investigated. In transmission electron microscopy (TEM), we observed the elongated morphology of diblock conjugated CP, with an average length of 210 nm calculated for diblock conjugates (**3**) prepared in water at 1 mg mL$^{-1}$.

It was not possible to observe the fully hydrophilic control conjugates (**4**) in TEM due to their size (around 12 nm in SLS, *vide infra*) and the highly dynamic nature of the assemblies of hydrophilic CP-polymer nanotubes, as expected from our previous work[16,22,38]. Small angle neutron scattering (SANS) was used to corroborate the nanotubular morphology of the peptide nanotubes in solution. Using SASfit software, the scattering profile of the diblock (**3**) and control (**4**) conjugates in D$_2$O were best fitted to a cylindrical micellar model. The resulting fits showed the average length to be over 100 nm, (the size window available to SANS analysis) for the diblock conjugate (**3**), and 10 nm for the control conjugates (**4**) (Fig. 2c). A detailed

analysis of the SANS fitting and experimental is given in Methods section and Supplementary discussion.

The aggregate length and the number of participating unimers of the control conjugate (**4**) were determined using static light scattering (SLS) as 12 nm and 25 nm, respectively. The aggregation number ($N_{agg}$) was calculated by dividing the average molecular weight of the aggregate, obtained by SLS, with the molecular weight of the unimer. The theoretical spacing of 0.5 nm between the cyclic peptides[46] in the nanotubes was used to extrapolate the average length of the nanotubes (i.e. $N_{agg} \times 0.5$ nm). The nanotubular length of the diblock conjugate **3** solutions prepared at 1 mg mL$^{-1}$ were consistently higher than the control conjugates (**4**) (see Fig. 2c and the Supplementary Figs. 2–4) for details.

The fluorescent dye, 1,6-diphenylhexatriene (DPH)[47], which is quenched in water, was employed to confirm the hydrophobic region at the core of the nanotubes, as depicted in Supplementary Fig. 1. Supplementary Fig. 1 shows the fluorescence of the DPH observed for the diblock conjugate (**3**), but not for the control conjugate (**4**), which does not have a hydrophobic region.

To study the stability of these nanotubes we determined their average size via SLS, after subjecting them to sonication for 1 h. The samples were then left to stand and SLS was remeasured both after 1 h and after 3 days. In a dynamic system, the reduction of $N_{agg}$, caused by the sonication will be corrected if the systems if left to equilibrate. This would result in a return to the original, and most thermodynamically stable, aggregation number. In a kinetically trapped, non-dynamic system, any decrease in the $N_{agg}$ due to sonication will remain relatively unchanged over time, due to the high stability of the assemblies. Therefore, the difference in tubular length between the original assemblies and the sonicated structures can be used as a measure for kinetic lability of CPNT-polymer conjugates.

As visible in Fig. 3, the aggregation number of the diblock conjugates (**3**) was significantly reduced after sonication across all concentrations. Indeed, at 1 mg mL$^{-1}$, the average length of the original nanotubes was found to be 273 nm while, after sonication, values of 94 nm and 89 nm were measured after

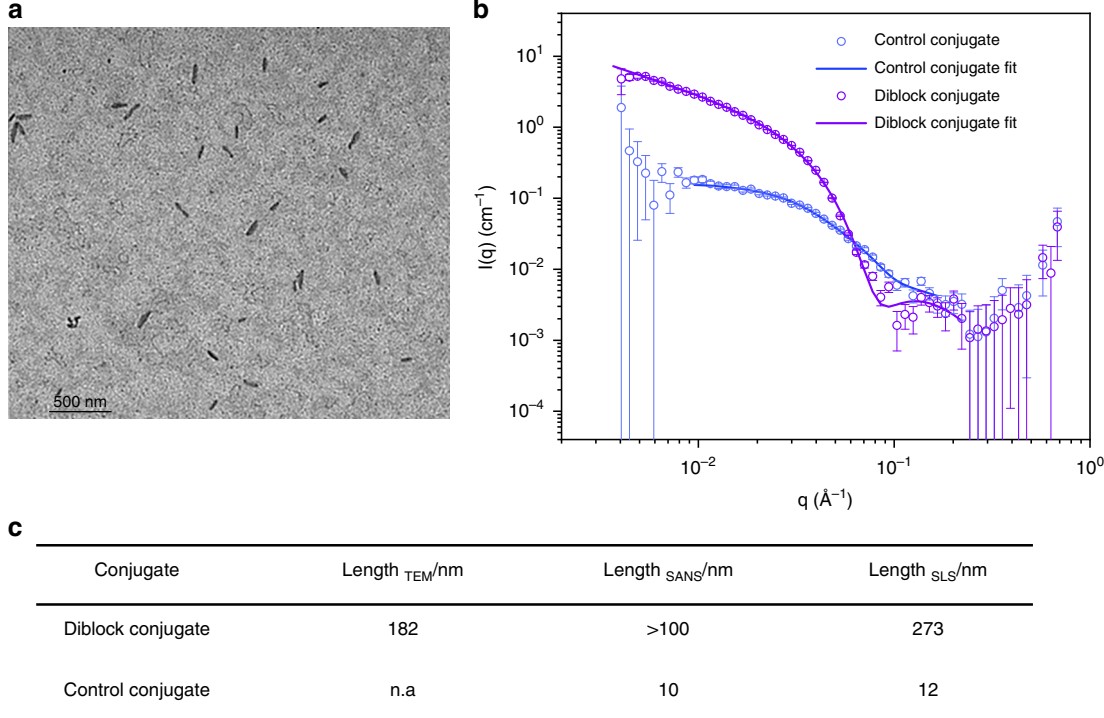

| Conjugate | Length $_{TEM}$/nm | Length $_{SANS}$/nm | Length $_{SLS}$/nm |
|---|---|---|---|
| Diblock conjugate | 182 | >100 | 273 |
| Control conjugate | n.a | 10 | 12 |

**Fig. 2** Characterisation of cyclic peptide-polymer nanotubes. **a** Transmission electron microscopy (TEM) of the diblock conjugates prepared in water. **b** Small-Angle Neutron Scattering (SANS) of the diblock (**12**) and control (**13**) conjugates fitted to a cylindrical micelle model using SASfit software. The error bars plotted are standard error of the mean. **c** Table of the average length of the nanotubes measured via TEM, SANS and static light scattering (SLS), respectively

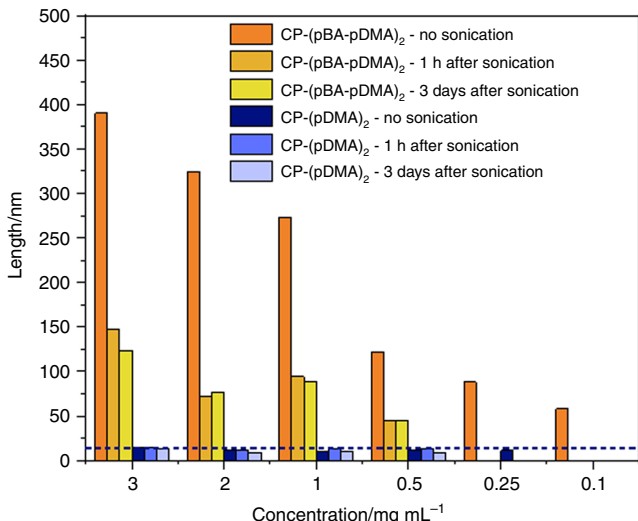

**Fig. 3** Dynamics of cyclic peptide-polymer nanotubes. The calculated length of the nanotubes from static light scattering over a range of concentrations (0.1–3.0 mg mL$^{-1}$) before and after sonication

being left to stand for 1 h and 3 days, respectively. Conversely, no change was observed for the control conjugates (**4**), which were 11, 14, 10 nm before and after sonication (1 h and 3 days after they were left to stand)—Supplementary Table 2. This corroborates our hypothesis that the hydrophobic region helps to stabilise the peptide self-assembly and reduces the dynamic nature of these systems[39]. In contrast, the hydrophilic control conjugates which are highly dynamic are able to equilibrate rapidly to their original thermodynamically favourable state, explaining why no net change in $N_{agg}$ is observed.

**FRET study**. To provide further insight into the dynamic behaviour of these supramolecular assemblies, Förster resonance energy transfer (FRET) dyes were conjugated to the periphery of the cyclic peptide-polymer nanotubes, to observe the rates of exchange of these systems. Using orthogonal amine protecting groups, selective deprotection of the amine was used to first attach the dye, then polymers to the peptide[23]. A detailed synthetic protocol and characterisation are given in the method section and Supplementary Figs. 6, 7, 22.

Cyanine (Cy) 3 (FRET donor) and Cy5 (FRET acceptor) were chosen as the FRET pair to detect the mixing of cyclic peptide conjugate unimers between nanotubes, see Fig. 4a. If the Cy3 and Cy5 dyes are in close in proximity to one another, upon excitation of the donor dye, we should observe the emission of the acceptor dye due to energy transfer between the FRET pair[48]. This proximity-dependent energy transfer can only take place when two different dye conjugates are assembled together in the same nanotube (each cyclic peptide is around 7.5 Å in diameter and the distance between the two peptides is 4.5 Å; the FRET range is between 10 to 100 Å), see Fig. 4b, c[49,50]. A control study with free dyes shows a constant FRET ratio (0.14) and no change in donor or acceptor emission over time is observed (Fig. 4d). Each unimer contains one dye molecule (either Cy3 or Cy5). Independently, Cy3 and Cy5 conjugates were pre-assembled in water and then co-injected together. If the cyclic peptides are highly dynamic, i.e., they disassemble and re-assemble readily, they form progressively mixed nanotubes. As donor and acceptor dyes come in closer proximity with every exchange, the FRET emission increases over time. This process should eventually lead to a statistically mixed nanotube where the Cy3 and Cy5 modified peptide should be randomly distributed throughout the aggregate and a constant final FRET ratio is reached.

Notably, in the control dye conjugate (**7** and **8**) the increase in FRET ratio was very fast and the plateau (no net change in FRET ratio) was reached within 60 min—as expected for a highly

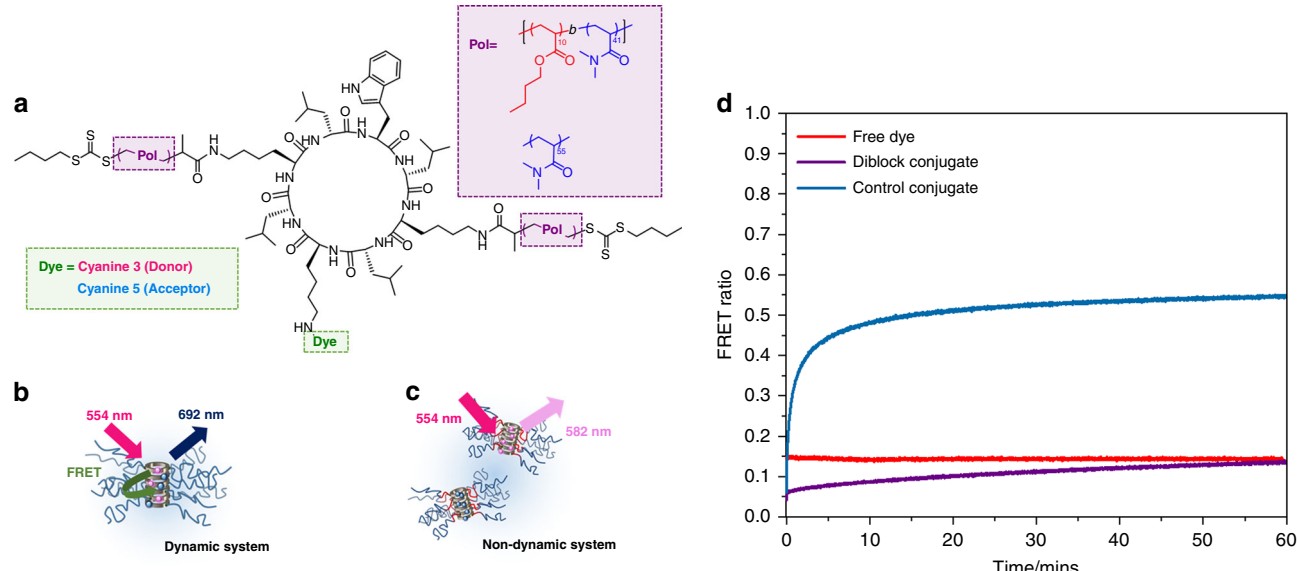

**Fig. 4** Kinetics of cyclic peptide-polymer nanotubes. **a** The orthogonal functionalization of cyclic peptide (CP) with two polymer arms and a donor or acceptor FRET dye. **b** Schematic of stable non-exchanging CP-polymer nanotubes. **c** Scheme of dynamically exchanging mixed CP-polymer nanotubes. **d** Graph to show the change in FRET ratio over time measured using fluorescence spectroscopy

**Table 1 Rate constants for FRET exchange, final and maximum FRET ratio and degree of mixing for the free dye, diblock (5 and 6) and control (7 and 8) dye conjugates**

| Sample | $k_1/s^{-1}$ | $k_2/s^{-1}$ | Final FRET ratio | Maximum FRET ratio | Degree of mixing/% |
|---|---|---|---|---|---|
| Free dye | 0 | 0 | 0.14 | 0.14 | – |
| Diblock conjugate | 0.0961 | 0.314 | 0.39 | 0.96 | 41 |
| Control conjugate | 0.0272 | 5.35 | 0.54 | 0.61 | 89 |

Rate constants were determined by fitting to a second order decay function, see Supplementary Fig. 8. Final FRET ratio was reached for the diblock and control conjugates after 7 days and 60 min, respectively

dynamic system. In contrast, the change in FRET ratio is extremely slow, when the diblock dye conjugates (**5** and **6**) are co-injected. To elucidate a maximum for the FRET ratio for both systems, another control experiment was set up. The conjugates were premixed in DMF, where they primarily exist as unimers. The DMF was removed and the conjugates were resuspended in water to measure the maximum FRET ratio of statically mixed nanotubes, see Table 1. A percentage degree of mixing was calculated by comparing the final FRET ratio (at the plateau) to the maximum FRET ratio from statistically mixed nanotubes. The degree of mixing for the control dye conjugates (**7** and **8**), as with previous hydrophilic CP-polymers without a stabilising hydrophobic core block, was around 90% or higher, indicating dynamic exchange of subunits to form statically mixed aggregates[23]. Also the diblock conjugate (**5** and **6**) after 7 days reached a final degree of mixing of only 41%, which suggests that the nanotubes are not fully mixing and discrete sections in the supramolecular assembly remain unchanged. The exchange rates could be fit to a second order decay function relating to the monomer association/dissociation and nanotube coagulation/fragmentation[23,51]. The significant retardation in the rate of exchange ($K_1$ and $K_2$), provides further evidence the hydrophobic pBA core-block stabilises the hydrogen bonded peptide self-assembly; affording a much less dynamic system.

**STORM characterisation**. To directly visualise the assembled peptide aggregates and any exchange between aggregates we imaged the nanotubes using super-resolution fluorescence microscopy, specifically direct stochastic optical reconstruction

microscopy (STORM)[52], see Supplementary methods for details. Representative STORM images are shown in Fig. 5 and also in the Supplementary Figs. 10, 11. In general the aggregates were well suited to imaging using STORM and the resolution of the final images were ~10–20 nm as calculated using the Fourier ring correlation method[53]. The images were recorded consecutively using a single laser excitation for each respective dye, the laser was set to 647 nm for Cy5 and then 568 nm for the Cy3. The co-injected and premixed conjugates were prepared using the same method as the FRET ratio study (*vide supra*). The general size and shape of the aggregates seemed similar. In the premixed samples the Cy3 and Cy5 dyes were significantly co-localised, suggesting the aggregates were all formed from combinations of Cy3 and Cy5 labelled peptides. In contrast, the co-injected Cy3 and Cy5 diblock conjugates, left to mix after 1 day, featured much more aggregates which were just a single colour, showing that the Cy3 and Cy5 dyes were not as well mixed as in the premixed sample.

The stability of the hydrophobic core prevents the free exchange of CP-conjugates between nanotubes, explaining why the Cy3 and Cy5 cannot be seen randomly distributed throughout the nanotubes as is the case for the premixed sample. Moreover, in several cases block-like structures can be observed where two conjugates with different colours were attached to each other. These observations demonstrate that the ends of the nanotubes are still able to assemble, to form supramolecular block co-polymers. As a result some degree of FRET exchange was observed, but the degree of mixing remains still below 50% even after 7 days. Meijer and coworkers recently showed a similar phenomenon in organic solvents[54].

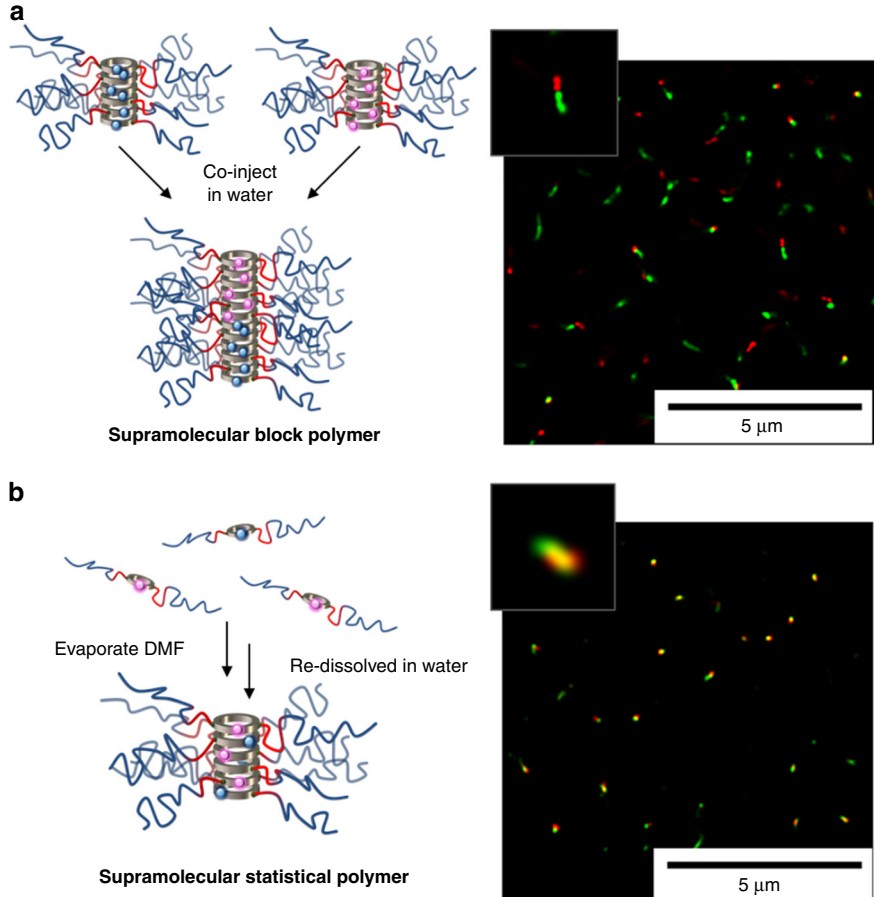

**Fig. 5** Composition of cyclic peptide-polymer nanotubes. Schematic and stochastic optical reconstruction microscopy (STORM) of **a** co-injected and **b** premixed cyclic peptide-polymer-dye conjugates

Due to the wavelength of the SLS laser overlapping with the dye on the FRET conjugates, another method was required to observe the average size of the aggregates. Multiple particle tracking was performed to work out an average hydrodynamic volume of the different dye conjugates and their mixtures, see Supplementary Table 4, Supplementary Fig. 9 and Supplementary Methods. Most importantly, the co-injected and premixed mixtures showed similar hydrodynamic radii (~50 nm), see Supplementary Table 4. It is worth noting that this is not a nanotube length, as the model assumes a spherical structure. The results nevertheless show that there are no significant differences in the average size between the aggregates. Even though the particle tracking analysis shows no significant differences in size, the STORM images of the premixed systems look slightly shorter this could be due some residual DMF disrupting the aggregation.

As previously discussed, a clear application of these cyclic peptide nanotubes are their use in drug delivery. To benefit from supramolecular system, the ability to form stabilised particles using the diblock polymer is a highly desirable design feature. Previously, the biocompatibility of self-assembling cyclic peptide-polymer nanotubes, with hydrophilic polymers, were tested and showed no change to cell viability[16]. To scope out the potential use of these diblock systems for biological applications and ensure the introduction of the hydrophobic region does not adversely affect the biocompatibility the diblock and control conjugates were tested for toxicity in cells using the XTT assay. PC-3 cells (epithelial prostate cancer cells) were treated over 24 h with the conjugates. The assay was used to examine the viability of the cells in the presence of the nanotubes. Both diblock (**3**) and

control (**4**) conjugate showed no toxicity (0.1–100 μM, see Supplementary Fig. 5).

In summary, we report a generation of supramolecular cylindrical nanostructures which are stable, bio-compatible and easily functionalised. Unlike previous hydrophilic cyclic peptide conjugates which show fast dynamics and a low aspect ratio, by introducing a secondary hydrophobic driving force to stabilise the peptide assembly we observe the formation of a stable supramolecular polymer with a length above 100 nm. Furthermore, controlling the dynamic behaviour using additional hydrophobic interactions enables the formation of supramolecular block co-polymer structures. With these experiments we prove that these highly functional supramolecular polymer brushes are able to form defined nanostructures similar to previously reported CDSA materials but, importantly for biological applications, in aqueous conditions.

## Methods

**Materials**. Fmoc-protected amino acids and coupling agents were purchased from Iris Biotech GmbH. Cyanine3 NHS ester and Cyanine5 NHS ester were obtained from Lumiprobe GmbH, Germany. The RAFT agent, (propanoic acid)yl butyl trithiocarbonate (PABTC), was synthesised in our group using literature protocol.[50] The initiator V-601 was purchased from Fujifilm Wako Pure Chemical Corporation. N-N-Dimethylacryliamide (DMA, Sigma-Aldrich, 99%) and butyl acrylate (BA) were filtered through a basic aluminium oxide (Fisher Scientific) column to deinhibit the monomer to remove the radical inhibitor before polymerisation reactions. All other chemicals stated were purchased from Sigma-Aldrich now Merck, (Gillingham, UK) unless otherwise stated. Solvents were purchased from several departmental suppliers—Honeywell, Fisher and Sigma-Aldrich.

**Nuclear magnetic resonance (NMR) spectroscopy.** [1]H NMR spectra were measured using a Bruker DPX-300 or DPX-400 NMR spectrometer, which operated at 300.13 MHz and 400.05 MHz, respectively. The residual solvent peaks were used as internal references. The following deuterated solvents were used: chloroform-d (CDCl₃), dimethyl sulfoxide-d6, deuterium oxide (D₂O). Chemical shift values (δ) are reported in ppm. The residual proton signal of the solvent was used as internal standard.

**Size exclusion chromatography (SEC).** Molar mass distributions were obtained using size exclusion chromatography (SEC). The polymers and conjugates were measured using the Agilent 390-LC MDS instrument which measured differential refractive index (DRI). Samples were prepared around 1–3 mg mL⁻¹ and filtered using 0.2 µM PTFE filters before auto-sampler injections. SEC THF: Agilent 1260 Infinity II-MDS, THF with 2% TEA + 0.01% BHT, RI, Viscometer, LS, MWD, 2× PLgel Mixed-C. SEC DMF: Agilent 1260 Infinity II-MDS, DMF with 5 mM NH₄BF₄, RI, Viscometer, VWD, LS, 2× PLgel Mixed-D. SEC DMF LiBr: Agilent PL50, DMF + 0.1% LiBr, DRi, UV, 2*Polargel M Columns.

**High-performance liquid chromatography (HPLC).** High-performance liquid chromatograms were measured using an Agilent Technologies 1200 Series, equipped with a Luna 5u C18 100 Å, 250 mm × 4.6 mm column. Acetonitrile/water was used. All solvents contained 0.04 vol% TFA. The following method was used: 0–5 mins 5:95 (Acetonitrile: Water), 5–35 mins gradient from 5:95 to 95:5 (Acetonitrile: Water), 35–38 mins gradient from 95:5 to 5:95 (Acetonitrile: Water).

**Ultraviolet–visible (UV–vis) absorption spectroscopy.** Spectroscopy: UV–vis absorption spectra were measured using an Agilent Technologies Cary 60 UV–vis spectrometer. The solutions were all made to the same concentration (35 µM).

**Fluorescence emission spectroscopy.** Fluorescence emission spectra were measured using an Agilent Technologies Cary Eclipse Fluorescence spectrometer. The FRET studies were performed using an absorption maxima of the donor conjugate (measured in using UV-Vis spectroscopy).

**Transmission electron microscopy (TEM).** Carbon coated grids, carbon film on copper 300 mesh, were purchased from EM Resolutions. Solutions of CP-diblock conjugate in water at 1 mg mL⁻¹ were prepared by direct dissolution of the solid in filtered ultra-pure water. Ten microlitre of solution was dropcast on freshly glow-discharged carbon-coated grids placed on filter paper. Bright field TEM micrographs were obtained with a Jeol 2100Plus is operating at 200 kV, equipped with a Gatan OneView IS camera.

**Static light scattering (SLS).** The SLS data were obtained using the ALV/CGS-3 Compact Gonimeter System. A range of different concentrations of the diblock and control conjugates were measured using SLS, before and after sonication. The compounds were first accurately measured in a vial and corresponding volumes of water were added to yield various different concentrations (0.5–3 mg mL⁻¹). Before measuring, the solutions were filtered using 0.45 × 10⁻⁶ m membrane PTFE lined filters to remove any large aggregates.

**Small-angle neutron scattering (SANS).** was carried out on the LARMOR small-angle diffractometer at the ISIS Pulsed Neutron Source (STFC Rutherford Appleton Laboratory, Didcot, UK). Prior to measurement, each sample was dissolved in D₂O and placed in a 2 mm quartz cuvette. The scattering cross-section was measured over a Q-range of 0.004–0.5 Å⁻¹ where Q is defined as:

$$Q = \frac{4\pi\sin\frac{\theta}{2}}{\lambda} \tag{1}$$

Here, $\theta$ is the scattered angle, and $\lambda$ is the incident neutron wavelength. A Q-range of 0.004–0.5 Å⁻¹ was achieved utilising an incident wavelength range of 0.9–13.3 Å. The detector is located 4.1 m from the sample and is 664 mm wide × 664 mm high with the beam in the centre of the detector. The beam size is 6 mm wide and 8 mm high. Each raw scattering data set was corrected for the detector efficiencies, sample transmission and background scattering and converted to scattering cross-section data ($\partial\Sigma/\partial\Omega$ vs. Q) using the instrument-specific software. These data were placed on an absolute scale (cm⁻¹) using the scattering from a standard sample (a solid blend of hydrogenous and perdeuterated polystyrene) in accordance with established procedures.

Following data acquisition, the SASfit software programme was used to model the data. In all cases several fixed parameters were used; concentration, the radius of the core, and SLD values (for the solvent, core peptide, and polymer shell).

Values for SLD were calculated using the following Eq. (2);

$$\text{SLD} = \frac{\rho N_a \sum_{i=1}^{N} b_i}{\sum_{i=1}^{N} M_i} \tag{2}$$

where $\rho$ is the bulk density of the material, $N_a$ is Avogadro's constant, $b_i$ is the scattering length contributions from the $N$ atoms within the unit cell, and $M$ is the

atomic mass of $N$ atoms. Values for individual atomic scattering lengths and atomic weights were taken from the NIST database[3,4]. SLD parameters were calculated for the solvent, polymer, and peptide and used as fixed values in the fitting analysis. The use of data from previous SANS experiments provided the vale for cylinder radius of 0.4 nm.

The hairy-cylinder model, is a combination model adapted from the form factor of a micelle, with a rod-like core as an additionally structure factor. The form factor can be described as;

$$P(q) = N^2\beta_s^2 F_{s(q)} + N\beta_c^2 F_{c(q)} + 2N^2\beta_s\beta_c S_{sc(q)} + N(N-1)\beta_c^2 S_{cc(q)} \tag{3}$$

where $N$ is the aggregation number, and $\beta_s = V_s(\rho_s-\rho_{solv})/\beta_c = V_c(\rho_c-\rho_{solv})$ is the total excess scattering lengths of the cylindrical core and in the corona, respectively. $V_s$ and $V_c$ are the volumes of the core and in the corona, respectively. $\rho_s$ and $\rho_c$ are the corresponding scattering length densities and $\rho_{solv}$ is the scattering length density of the surrounding solvent, calculated as previously discussed. The solutions were prepared by dissolving the conjugates in 1 mg mL⁻¹ of D₂O.

**1,6-diphenylhexatriene (DPH) dye experiment.** DPH was dissolved to make a 1 mg mL⁻¹ solution in DMSO. Under the UV lamp (365 nm) the dye fluorescence was clearly observed. In separate vials, accurately prepared 1 mg mL⁻¹ solutions of the diblock and control conjugates solutions were made up in de-ionised water. The solutions were left on a shaker overnight to full dissolve and equilibrate. Before dye addition the fluorescence of the de-ionised water and DMSO (solvent only) and the conjugates in solution were checked. Using a micro-pipette exactly 5% vol ratio of the dye solution in DMSO was added to the conjugates.

The fluorescence of DPH quenches in the presence of water. As the DPH dye enters a hydrophobic region where it can shield itself from the water, fluorescence can once again be observed. When the dye enters the hydrophobic pBA core around the cyclic peptide, the dye fluorescence can be observed. Clear fluorescence emission in the diblock conjugate solution was observed after 10 mins. No fluorescence was observed in the solvent or the control conjugate solution after 3 days.

**Toxicity: cell culture.** PC-3 (epithelial prostate cancer cells) cells were grown in Dulbecco's modified Eagle's medium (DMEM) supplemented with 10% (v/v) foetal bovine serum (FBS) and 2 mM of L-glutamine and penicillin at 37 °C in a humid 5% CO₂ environment. Cells were typically passaged at 80–90% confluence.

**Cytotoxicity assay (XTT/PMS).** Toxicity of the conjugates was assessed using a standard XTT protocol. The CP conjugates to be tested were dissolved in water with 0.5% DMSO in order to obtain solutions at 500 µM. These solutions were used to prepare dilutions in a mixture of supplemented culture media (DMEM) and PBS (50:50) at the following concentrations: 100, 50, 10, 1 and 0.1 µM. PC-3 cells were seeded in a transparent Greiner 96 well-plate at a density of 10,000 cells per well and incubated for 24 h. The culture media was then replaced by 100 µL of the prepared solutions. After 24 h incubation, the medium was removed, the cells were washed once with PBS before adding fresh media supplemented with 25 µL of XTT solution (1 mg·mL⁻¹) containing N-methyl dibenzopyrazine methyl sulfate (PMS) (25 µmol.L⁻¹). Cells were incubated for another 24 h. Absorbance was then directly measured using a BioTek™ Cytation™ 3 Cell Imaging Multi-Mode Reader at 450 nm and 650 nm (background). Two repeats of this experiment were performed; the error bars plotted are standard error of the mean.

**FRET exchange study.** Solutions of Cy3 and Cy5 conjugates were prepared in water at 35 µM. The FRET ratio over time was measured, after co-injection of the dye conjugate pair (1:1 vol ratio), by fluorescence emission microscopy. Upon excitation at the donor absorption maxima, the emission at donor and acceptor maxima of the dyes were used to calculate the FRET ratio. The energy transfer from the donor to the acceptor i.e. the FRET measured, can be related to the exchange of cyclic peptides between nanotubes in solution. Furthermore the change in FRET over time can be modelled to a second order decay function[5,6], to calculate the rate of exchange taking place. The fitting below was done using OriginPro software. Details of the fit can be found below.

**Polymer synthesis.** pBA (**17**): For the synthesis of the first block, PABTC (0.214 g, 0.900 mmol, 1 eq.), butyl acrylate (1.153 g, 9.00 mmol, 10 eq.), V601 azo-initiator (4.14 mg, 0.018 mmol, 0.02 eq.) and 1,4-dioxane (0.901 g) were all weighed into a vial with a magnetic stirrer and sealed with a rubber septum. The solution was mixed thoroughly and deoxygenated by bubbling nitrogen for ca. 10 min. The vial was then placed in an oil bath set at 70 °C for 20 h. Samples conversion, calculated from [1]H NMR, during the polymerisation were taken using a degassed syringe. After the polymerisation, the mixture was cooled and opened to air. [1]H NMR and GPC of these polymers were taken to determined, conversion and molecular weight.

Yield was not calculated as the solution containing the macroCTA was used directly to synthesis the following diblock. [1]H NMR (CDCl₃, 300 MHz, δ ppm): 0.94 (3 H, **H₃C**–CH₂–RAFT agent + 3 H, **H₃C**–CH₂–BA monomer), 1.39 (2 H, H₃C–**CH₂**–BA monomer), 1.61 (2 H, H₃C–CH₂–**CH₂**–BA monomer), 3.35 (–**CH₂**–S–RAFT

agent), 4.05 (2 H, –**CH$_2$**–O–C(O)–BA monomer). $M_n = 1760$ g mol$^{-1}$, Đ = 1.15 (THF SEC, Agilent EasyVial PMMA calibration). The $^1$H NMR spectrum (Supplementary Fig. 12) and SEC chromatogram (Supplementary Fig. 15).

pBA-*b*-pDMA (**1**): An exact aliquot of the mixture containing the first block (pBA) was taken by calculating the moles of MacroCTA (pBA-block) from the known concentration of your pBA polymerisation. In a new vial, the aliquot containing your pBA-block (900 mg, 0.254 mmol, 1 eq.), N,N-Dimethylacrylamide (DMA) (1.134 g, 11.43 mmol, 45 eq.), V-601 (0.84 mg, 0.0143 eq.) and 1,4-Dioxane (1.382 g) were added together. The solution was mixed thoroughly and deoxygenated by bubbling nitrogen for ca. 10 min. The vial was then placed in an oil bath set at 70 °C for 20 h. The solution was precipitated in diethyl ether and dried in a vacuum oven. The product was a yellow solid.

Yield = 85% (1.2292 g); $^1$H NMR (CDCl$_3$, 300 MHz, ppm): 0.94 (3 H, **H$_3$C**–CH$_2$–RAFT agent + 3 H, **H$_3$C**–CH$_2$–BA monomer), 1.37 (2 H, H$_3$C–**CH$_2$**–BA monomer), 1.61 (2 H, H$_3$C–CH$_2$–**CH$_2$**–BA monomer), 2.91 (6 H, –N–(**CH$_3$**)$_2$), 3.34 (–CH$_2$–S– RAFT agent), 4.03 (2 H, –**CH$_2$**–O–C(O)–BA monomer). $M_n = 5000$ g mol$^{-1}$, Đ = 1.13 (THF SEC, Agilent EasyVial PMMA calibration). The $^1$H NMR spectrum (Supplementary Fig. 13) and SEC chromatogram (Supplementary Fig. 15).

pDMA (**2**): For the synthesis of the pDMA homopolymer, PABTC (50.06 mg, 0.210 mmol, 1 eq.), DMA (1.041 g, 10.50 mmol, 50 eq.), VA-044 azo-initiator (1.49 mg, 4.60 μmol, 0.0219 eq.) and a 1:4 co-solvent of 1,4-dioxane (0.421 mL) and deionised water (1.264 mL) respectively were all weighed into a vial with a magnetic stirrer and sealed with a rubber septum. The solution was mixed thoroughly and deoxygenated by bubbling nitrogen for *ca.* 10 min. The vial was then placed in an oil bath set at 70 °C for 20 h. Samples conversion, calculated from $^1$H NMR, during the polymerisation were taken using a degassed syringe. After the polymerisation, the mixture was cooled and opened to air. $^1$H NMR and GPC of these polymers were taken to determined, conversion and molecular weight. The solvent was evaporated using the aid of nitrogen flow, then the polymer was resuspended in dioxane and precipitated in hexane (repeat 3 times) and dried in a vacuum oven. The product was a yellow solid.

Yield = 88% (1.0483 g); $^1$H NMR (CDCl$_3$, 300 MHz, ppm): 0.89 (3 H, **H$_3$C**–CH$_2$–RAFT agent, 1.08 (3 H, **H$_3$C**–(C(O)–OH)–RAFT agent), 1.85–1.46 (4 H, H$_3$C–**CH$_2$**–**CH$_2$**–RAFT agent + 2 H, –CH–**CH$_2$**– polymer backbone), 2.62 (–**CH**–CH$_2$– polymer backbone), 2.80–3.25 (6 H, –N–(**CH$_3$**)$_2$). $M_n = 5000$ g mol$^{-1}$, Đ = 1.13 (THF SEC, Agilent EasyVial PMMA and PS calibration). The $^1$H NMR spectrum (Supplementary Fig. 14) and SEC chromatogram (Supplementary Fig. 17).

**Cyclic peptide-polymer conjugates**. CP-(pBA-pDMA)$_2$ (**3**): The cyclic peptide (**10**), cyclo(D-Leu-Lys-D-Leu-Trp)$_2$, was synthesised using literature protocol[1]. CP (15 mg, 13.88 μmol, 2.2 eq.) was dissolved in DMF (0.5 mL) with the aid of sonication. In a separate vial, pBA-pDMA (**1**) (0.171 g, 30.54 μmol, 2.2 eq.), HATU (0.0116 g, 30.54 μmol, 2.2 eq.) and DIPEA (0.0108 g, 83.28 μmol, 6 eq.) were dissolved in 0.5 mL DMF and shaken for 30 min then added to the CP solution. The combined solution was shaken for 2 days at room temperature. The product then precipitated in diethyl ether five times. To ensure the polymer stays in solution and is removed with the supernatant during the precipitation procedure a small amount of methanol used to redissolved both polymer and conjugate before precipitation solvent, diethyl ether, was reintroduced. A minimum amount of diethyl ether was used to precipitate the conjugate. The purification of the excess polymer can be monitored via SEC traces (Supplementary Fig. 16). Yield = 47% (84.5 mg).

CP-(pDMA)$_2$ (**4**): CP (15 mg, 13.88 μmol, 2.2 eq.) was dissolved in DMF (0.5 mL) with the aid of sonication. In a separate vial, pDMA (**2**) (0.174 g, 30.54 μmol, 2.2 eq.), HATU (0.0116 g, 30.54 μmol, 2.2 eq.) and DIPEA (0.0108 g, 83.28 μmol, 6 eq.) were dissolved in 0.5 mL DMF and shaken for 30 min then added to the CP solution. The combined solution was shaken for 2 days at room temperature. The solvent was evaporated with the aid of a nitrogen flow. The product then purified using centrifuge dialysis tubes with a molecular weight cut-off of 10 kDa (Merck Millipore). The purification of the excess polymer can be monitored via SEC traces (see Supplementary Fig. 18). Yield = 41% (71.5 mg).

**Cyclic peptide-polymer-dye conjugates**. Cy3-CP-protected (**13**): The partially deprotected cyclic peptide was synthesised according literature protocol[2]. This cyclic peptide was dissolved in 0.5 mL of DMF with the aid of sonication. N,N-Diisopropylethylamine (DIPEA) (0.0117 g, 90.7 μmol, 6 eq.) was added to the CP solution and mixed. Cyanine3 NHS ester (purchased from Lumiprobe GmbH) (0.011 g, 17.4 μmol, 1.15 eq.) was added to the CP solution and stirred for 3 days. The reaction was followed via HPLC, Supplementary Fig. 20. The purified peptides were characterised by mass spectrometry (electrospray ionisation, ESI) (Supplementary Table 1). Yield: 78% (20.6 mg).

Cy3-CP-deprotected (**14**): Boc groups were removed in using a deprotection solution of TFA/TIPS/H2O (18:1:1 vol, 5 mL). The dye conjugated Boc protected CP (**13**) (20.632 g) was agitated for 3 h in the deprotection solution, then triturated using ice-cold diethyl ether and washed twice more with ice-cold diethyl ether. The pink precipitate was collected and dried under vacuum. The reaction was followed via HPLC, Supplementary Fig. 20. Mass spectra attributions can be found in Supplementary Table 1. Yield: 94% (25.7 mg).

The Cyannine5 conjugates (**15** and **16**) were synthesised using the same procedure to synthesise Cyannine3 conjugates. The reaction was followed via HPLC, Supplementary Fig. 21. Mass spectra attributions can be found in Supplementary Table 1. Cy5-CP-Protected (**15**): Yield: 92% (25.4 mg). Cy5-CP-Deprotected (**16**): Yield: 65% (13.7 mg).

Cy3-CP-(pBA-pDMA)$_2$ (**5**): Cyclic peptide-dye conjugate (Cy3-CP-dep, **14**) (16 mg, 10.9 μmol, 1 eq.) was dissolved in DMF (0.5 mL) with the aid of sonication. In a separate vial, pBA-pDMA (**1**) (0.138 g, 24.1 μmol, 2.2 eq.), HATU (9.2 mg, 24.1 μmol, 2.2 eq.) and DIPEA (8.5 mg, 65.7 μmol, 6 eq.) were dissolved in 0.5 mL DMF and shaken for 30 min then added to the CP solution. The combined solution was shaken for 3 days at room temperature. The product then precipitated in diethyl ether five times. To ensure the polymer stays in solution and is removed with the supernatant during the precipitation procedure a small amount of methanol used to redissolved both polymer and conjugate before precipitation solvent, diethyl ether, was reintroduced. A minimum amount of diethyl ether was used to precipitate the conjugate. The purification of the excess polymer can be monitored via SEC traces (see Supplementary Fig. 11). Yield = 43% (58.3 mg).

Cy3-CP-(pDMA)$_2$ (**7**): Cyclic peptide-dye conjugate (Cy3-CP-dep, **14**) (6.2 mg, 4.17 μmol, 1 eq.) was dissolved in DMF (0.5 mL) with the aid of sonication. In a separate vial, pDMA (**2**) (51.2 mg, 9.16 μmol, 2.2 eq.), HATU (3.48 mg, 9.16 μmol, 2.2 eq.) and DIPEA (3.23 mg, 25.0 μmol, 6 eq.) were dissolved in 0.5 mL DMF and shaken for 30 min then added to the CP solution. The combined solution was shaken for 3 days at room temperature. The solvent was evaporated with the aid of a nitrogen flow. The product then purified using centrifuge dialysis tubes with a molecular weight cut-off of 10 kDa (Merck Millipore). The purification of the excess polymer can be monitored via SEC traces (see Supplementary Fig. 11). Yield = 8% (4.390 mg).

The Cyannine5 conjugates (Cy5-CP-(pBA-pDMA)$_2$ (**6**) and Cy5-CP-(pDMA)$_2$ (**8**)) were synthesised using the same procedure to synthesise Cyannine3 conjugates (see Supplementary 22 for SEC).

## Data availability

All data that support the findings of the current study are available from the corresponding author upon reasonable request.

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

## Acknowledgements

The Royal Society Wolfson Merit Award (WM130055; S.P.), Monash-Warwick Alliance (S.P.), the European Research Council (TUSUPO 647106; J.Y.R.; E.M.; S.H.E.; R.P.; S.P.) and the German Research Foundation (DFG, GZ: HA 7725/1-1; M.H. and GZ: BR 4905/1-1, BR 4905/1-2; J.C.B.) are acknowledged for financial support. The authors acknowledge the Midlands Regional Cryo-EM Facility, hosted at the Warwick Advanced Bioimaging Research Technology Platform, for use of the JEOL 2100Plus, supported by MRC award reference MC_PC_17136, the University of Warwick Research Technology Platform for size exclusion chromatography (SEC) facilities. We would like to thank Robert Dalgleish for help on the Larmor in the Materials Characterisation Laboratory at the ISIS Neutron and Muon Source (RB1720086). The MRC funded the development of the super-resolution fluorescence microscopy (research grant EP/F062966/1).

## Author contributions

Design, synthesis and all characterisation unless otherwise stated was completed by J.R. STORM imaging and particle tracking analysis (H.C. and T.W), SANS fitting (E.M.), XTT assay (S.H.E. and R.P). Design and discussion (J.B, M.H and S.P).

## Competing interests

The authors declare no competing interests.
