## [Peer Review File · Nature Communications]

Reviewers' comments:

Reviewer #1 (Remarks to the Author):

The manuscript by Perrier and coworkers is focused on the hierarchical self-assembly of peptide nanotubes using alternating L-, D- cyclic octapeptide building blocks. This is an interesting topic, which goes beyond the simple process of self-assembly based on the interaction of a single type of building blocks. The work is technically sound and the conclusions are consistent with the experimental results.

The main limitation of the manuscript is the level of novelty as well as the description of the state-of-the-arts. A clear example of the narrow reporting of the scientific literature could be noticed when a 2008 paper is being cited as "Recent works have shown their use in wide range of applications in optoelectronics".

The authors also completely ignore the wide body of literature related to the hierarchical self-assembly of di- and tri-peptides as well as peptide amphiphiles. It is not truly clear while the alternating L-, D- cyclic octapeptide provides any advantage as compared to the much simpler systems.

On the technical side, the TEM characterization as presented in Figure 2a is extremely limited. A much higher resolution should be reported as well as the used of complementary methods.

Minor points:

1. Reference #28 page number should be 566-570.
2. Reference #39 is wrong. Should be "Sci. STKE"

Reviewer #2 (Remarks to the Author):

In this manuscript, authors reported that introducing a secondary hydrophobic interaction near the cyclic peptide core could stabilise the highly dynamic tubular structure and enable the formation of supramolecular block co-polymer structures in water. Experiments and characterizations were treated at very high standards, the phenomena and results observed are very impressive. Hence, Publication is recommended after consideration of the following minor suggestions:

1. As the term suggests, "hierarchical self-assembly" is a stepwise process in which components are brought together in a precisely defined way by non-covalent interactions. The units produced at each level, form the building blocks for self-assembly at the next higher level of complexity. From this point, in this manuscript, the self-assembly process is more similar to synergistic non-covalent interactions enhanced self-assembly. Therefore, the title is worth considering.
2. For the STORM, could the authors please provide the images under single laser excitation (for Cy3, Cy5 excitation) for co-injected and premixed sample respectively? Especially for Cy3 excitation, we can directly see if there is FRET effect by observing red (or yellow) fluorescence.
3. For the formation of supramolecular block copolymers, from STORM, the length of the block copolymers is bigger than premixed one, can the authors provide an explanation to this observation.

Reviewer #3 (Remarks to the Author):

The paper by Rho and co-workers describes a new self-assembly mode not known for cyclic peptides of alternating chirality. The work builds on the excellent seminal group by the Perrier group to stabilize individual cyclic peptide nanotubes in aqueous solution by using peptide/polymer hybrid architectures. This paper describes the hierarchically controlled self-assembly of cyclic peptide nanotubes by employing kinetically trapped tubular ensembles. The results are confirmed by a wide variety of spectroscopy and microscopy techniques (light scattering, FRET, STORM, etc.). Therefore, the high novelty of the work and the excellent and robust characterisation of the molecules described made this contribution suitable for publication.

The manuscript would benefit from improving the following points:

As the authors mention, the block-like hybrid tubes are nice and remarkable. However, the formation of supramolecular block polymers seems low yielding. It would be very interesting for future potential applications to grow them longer and with alternating blocks. The experimental results obtained suggested that despite the peptide/polymer hybrid architecture can strongly stabilize the tubular block, they might also inhibit the polymer growth by preventing block-to-block interactions. Doping these block mixtures with more dynamic, but also smaller and less hindered cyclic peptide connectors could help in growing longer and potentially more functional supramolecular crystals.

The paper is focused in the description of a new self-assembled architecture of cyclic peptide nanotubes. However, in the middle of the manuscript, just before the dynamic characterization (page 5), a cell viability assay is performed without any connection with the main message of the paper or any other further explanation. Either move this section to the end of the manuscript in line with potential future applications or directly remove it from the discussion.

Figure 3 shows the fragmentation of nanotubes after sonication. The authors claim that hydrophobic polymer bearing peptides nanotubes are unable to recover their original length distribution because they are kinetically trapped. On the other hand, the hydrophilic control does not show any length distribution difference after sonication. However, the length of the CP-(pDMA)₂ is too short to extract robust conclusions from this control. The concentration in weight per volume is not optimal for two molecules with very different molecular mass. Control experiments with nanotubes of similar length, perhaps playing with concentrations could help addressing this point.

In the introduction: "Many of these systems have been optimised to form very large elongated assemblies, typically nanofibers, which create supramolecular gel networks.^{3,8-10}" Please consider including this recent paper precisely about hydrogels by the controlled 1D elongation of cyclic peptide assemblies: pH-Triggered self-assembly and hydrogelation of cyclic peptide nanotubes confined in water micro-droplets. *Nanoscale Horizons*, 2018, 3(4), 391–396. <http://doi.org/10.1039/C8NH00009C>. Also in the introduction: "Recent works have shown their use in a wide range of applications, in optoelectronics,¹¹ self-healable materials,¹² or bio-therapeutics.¹³⁻¹⁵" Please consider including a recent review about functional supramolecular 1D assemblies: *Chemical Communications*, 2017, 53(56), 7861–7871. <http://doi.org/10.1039/C7CC02997G>.

Please define in the paper (or legend of the figure) and give a suitable reference(s) for the two rate constants for FRET exchange (k_1 and k_2) included in Fig. 4d.

The nomenclature for the compounds is not very clear. Ideally, all the structures should be specified in the figures.

No indication of the beta sheets is reported. Infrared could be used to confirm amide bonds compromised in antiparallel (or parallel) β -sheet supramolecular arrangement.

I found some minor typos including the Aida coranullene reference without Journal name (ref. 32).

Reviewers' comments:

For clarity the italic font has been used to denote any additions to the manuscript.

Reviewer #1 (Remarks to the Author):

The manuscript by Perrier and coworkers is focused on the hierarchical self-assembly of peptide nanotubes using alternating L-, D- cyclic octapeptide building blocks. This is an interesting topic, which goes beyond the simple process of self-assembly based on the interaction of a single type of building blocks. The work is technically sound and the conclusions are consistent with the experimental results.

The main limitation of the manuscript is the level of novelty as well as the description of the state-of-the-arts. A clear example of the narrow reporting of the scientific literature could be noticed when a 2008 paper is being cited as "Recent works have shown their use in wide range of applications in optoelectronics".

We agree with the reviewer's comment and the introduction has now been updated to better reflect the current body of work, with the following references added:

12. Jain, A. & George, S.J. *New directions in supramolecular electronics. Mater. Today* **18**, 206-214 (2015).

13. Jin, X.-H. et al. *Long-range exciton transport in conjugated polymer nanofibers prepared by seeded growth.* **360**, 897-900 (2018).

17. Fuertes, A., Juanes, M., Granja, J.R. & Montenegro, J. *Supramolecular functional assemblies: dynamic membrane transporters and peptide nanotubular composites. Chem. Commun.* **53**, 7861-7871 (2017).

The authors also completely ignore the wide body of literature related to the hierarchical self-assembly of di- and tri-peptides as well as peptide amphiphiles. It is not truly clear while the alternating L-, D- cyclic octapeptide provides any advantage as compared to the much simpler systems.

From the suggestion of the reviewers these examples have now been added. In particular, advantages of the cyclic peptide system over these systems have also been highlighted:

"As a motif to form supramolecular polymers, hydrogen bonded core-stacking moiety have gained much attention due to their ability to form directional nanostructures.^{1,4} The ease in functionalising these cores with a range of different substituents, such as polymers,⁵ dyes⁶ or drugs,⁷ make them ideal candidates for bioactive materials such as antimicrobials⁸ or drug delivery vectors.⁹ The most prominent example of hydrogen bonded self-assemblies are peptides. The amino acid building blocks of peptides provide an extensively library of functional groups whilst the amide backbone can prove an in-built self-assembly motif, which can take part in hydrogen bonding. The simplest example is the dipeptide system.^{10,11} Most notably, they have been shown to self-assembly to form networks of nanofibers, which have shown promising applications in tissue engineering¹² and drug delivery.¹³ Another notable example is the surfactant based peptide amphiphiles which contains a hydrophilic tail, β -sheet forming middle segment and an outer bio-active epitope.¹⁴⁻¹⁶ This two-fold self-assembly system has been used to develop bio-mimetic self-assembled nanofibers in order to aid bone mineralisation¹⁷ and wound healing.¹⁶

Among the systems reported to date, an exciting supramolecular candidate for potential biological applications is the family of self-assembling cyclic peptide nanotubes (CPNT). Alternating D- and L- octapeptide, upon cyclisation, have been shown to stack *via* hydrogen bonding to form a nanotubular assembly.^{18,19} *Due to the cyclic construction of the peptide this system also has more opportunity for functionalisation, compared to its linear counterparts. Previously, the multiple modification on the periphery of the cyclic peptide, either with different orthogonal polymers^{20,21} or multiple polymer arms (1-3)⁵ has been reported. Moreover, Xu and co-workers have shown that the interior of the cyclic peptide nanotubes can be tuned without compromising the ability to form nanotubular structures.²² The attachment of hydrophilic polymers to the periphery of the peptide core has been studied to provide water soluble aggregates.^{5,23,24} More recently, they have been shown to be pH and redox responsive,²⁵ have antimicrobial activity^{8,26} and be a promising drug carrier.^{7,9} "*

On the technical side, the TEM characterization as presented in Figure 2a is extremely limited. A much higher resolution should be reported as well as the used of complementary methods.

TEM at higher resolution and multiple images have now been included in the SI:

Magnification - 40,000

Magnification - 25,000

Magnification - 15,000

Magnification - 15,000

Attempts have been made to image these systems in SEM and AFM, however due to difficulty in sample preparation and the width of the aggregates (~10-20 nm) these methods struggled to obtain clear images of the nanotubes. However, note that the nanotubular length has been confirmed by both SLS and SANS. The SANS also confirms the nanotubular morphology of the aggregates in solution.

Minor points:

1. Reference #28 page number should be 566-570.
2. Reference #39 is wrong. Should be "Sci. STKE"

These changes have now been made.

Reviewer #2 (Remarks to the Author):

In this manuscript, authors reported that introducing a secondary hydrophobic interaction near the cyclic peptide core could stabilise the highly dynamic tubular structure and enable the formation of supramolecular block co-polymer structures in water. Experiments and characterizations were treated at very high standards, the phenomena and results observed are very impressive. Hence, Publication is recommended after consideration of the following minor suggestions:

1. As the term suggests, "hierarchical self-assembly" is a stepwise process in which components are brought together in a precisely defined way by non-covalent interactions. The units produced at each level, form the building blocks for self-assembly at the next higher level of complexity. From this point, in this manuscript, the self-assembly process is more similar to synergistic non-covalent interactions enhanced self-assembly. Therefore, the title is worth considering.

The referee has made a good point, the term hierarchical has been changed to 'dual' to denote two part process without implying stepwise:

"Dual self-assembly of supramolecular peptide nanotubes to provide stabilisation in water"

2. For the STORM, could the authors please provide the images under single laser excitation (for Cy3, Cy5 excitation) for co-injected and premixed sample respectively? Especially for Cy3 excitation, we can directly see if there is FRET effect by observing red (or yellow) fluorescence.

The images were recorded consecutively one laser at a time using a single laser excitation. The excitation of the laser has been set to a single excitation depending on the Cy3 or Cy5 excitation. See details in SI page 22. Following the reviewer's comment, we realize that this information could be easily missed, but is very important to the description of the experiment, so we have added a clarification in the main manuscript to make this more apparent:

"The images were recorded consecutively using a single laser excitation for each respective dye, the laser was set to 647 nm for Cy5 and then 568 nm for the Cy3."

SI page 22:

"To capture two colour images, two raw data sets were captured sequentially for each STORM image, one for the Cy5 dye (647 nm laser) and then another for the Cy3 dye (568 nm laser)."

3. For the formation of supramolecular block copolymers, from STORM, the length of the block copolymers is bigger than premixed one, can the authors provide an explanation to this observation.

The average size of the aggregates was calculated to be similar for both the co-injected and premixed systems using particle tracking analysis. The method used to premix the Cy3 and Cy5 dye conjugates requires mixing unimers in DMF, and residual DMF, which is very hard to remove, could be affecting the overall length of the aggregates upon re-dissolving in water. This comment has now been added to the manuscript:

"Even though the particle tracking analysis shows no significant differences in size, the STORM images of the premixed systems look slightly shorter this could be due some residual DMF disrupting the aggregation."

Reviewer #3 (Remarks to the Author):

The paper by Rho and co-workers describes a new self-assembly mode not known for cyclic peptides of alternating chirality. The work builds on the excellent seminal group by the Perrier group to stabilize individual cyclic peptide nanotubes in aqueous solution by using peptide/polymer hybrid architectures. This paper describes the hierarchically controlled self-assembly of cyclic peptide nanotubes by employing kinetically trapped tubular ensembles. The results are confirmed by a wide variety of spectroscopy and microscopy techniques (light scattering, FRET, STORM, etc.). Therefore, the high novelty of the work and the excellent and robust characterisation of the molecules described made this contribution suitable for publication.

The manuscript would benefit from improving the following points:

As the authors mention, the block-like hybrid tubes are nice and remarkable. However, the formation of supramolecular block polymers seems low yielding. It would be very interesting for future potential applications to grow them longer and with alternating blocks. The experimental results obtained suggested that despite the peptide/polymer hybrid architecture can strongly stabilize the tubular block, they might also inhibit the polymer growth by preventing block-to-block interactions. Doping these block mixtures with more dynamic, but also smaller and less hindered cyclic peptide connectors could help in growing longer and potentially more functional supramolecular crystals.

The paper is focused in the description of a new self-assembled architecture of cyclic peptide nanotubes. However, in the middle of the manuscript, just before the dynamic characterization (page 5), a cell viability assay is performed without any connection with the main message of the paper or any other further explanation. Either move this section to the end of the manuscript in line with potential future applications or directly remove it from the discussion.

The reviewer makes a very good point. As suggested, this section has been moved to the end and a sentence has been added to provide further context:

"As previously discussed, a clear application of these cyclic peptide nanotubes are their use in drug delivery. To benefit from supramolecular system, the ability to form stabilised particles using the diblock polymer is a highly desirable design feature. Previously, the biocompatibility of self-assembling cyclic peptide-polymer nanotubes, with hydrophilic polymers, were tested and showed no change to cell viability.¹⁶ To scope out the potential use of these diblock systems for biological applications and ensure the introduction of the hydrophobic region does not adversely affect the biocompatibility the diblock and control conjugates were tested for toxicity in cells using the XTT assay. PC-3 cells (epithelial prostate cancer cells)

were treated over 24 hours with the conjugates. The assay was used to examine the viability of the cells in the presence of the nanotubes. Both diblock (12) and control (13) conjugate showed no toxicity (0.1 – 100 μ M, see SI Figure S5).”

Figure 3 shows the fragmentation of nanotubes after sonication. The authors claim that hydrophobic polymer bearing peptides nanotubes are unable to recover their original length distribution because they are kinetically trapped. On the other hand, the hydrophilic control does not show any length distribution difference after sonication. However, the length of the CP-(pDMA)₂ is too short to extract robust conclusions from this control. The concentration in weight per volume is not optimal for two molecules with very different molecular mass. Control experiments with nanotubes of similar length, perhaps playing with concentrations could help addressing this point.

The aggregation of the control conjugate has been confirm in both SLS and SANS to be around 10 nm, equating to a number of aggregation of 30. Using both techniques, it is possible to differentiate a system that is assembled or disassembled. Using SANS, our group has previously studied systems that are mainly unimeric where there is little to no assembly - and this was not observed for the control conjugate.

The SANS data not only shows the size of the aggregates upon self-assembly but also shows that control conjugates fits better to an elongated cylindrical micelle than the spherical micelle model – confirming that even at these short nanotubular lengths an elongated core is notable. The spherical model fitting has now also been included in the SI:

Control conjugate - CYL+Chains (RW) - chisqr: 26.8, red. chisqr: 1.03

Parameters	Fit	Value	Units
Delta	✓	0.083845	
R_core		5	Å
n_agg	✓	0.00453034	
V_brush		15757	cm ³
eta_core		8.20995e-007	Å ⁻²
eta_brush		9.57751e-007	Å ⁻²
eta_solv		6.33e-006	Å ⁻²
xsolv_core	✓	0.572701	
Rg	✓	12.0219	Å
h		1	
H	✓	492.444	Å

Control conjugate - Sphere+Chains (RW) - chisqr: 1439.13, red. chisqr: 29.98

Parameters	Fit	Value	Units
Delta	✓	134.251	
R_core		5	Å
n_agg	✓	0.00500811	
V_brush		15757	cm ³
eta_core		8.20995e-007	Å ⁻²
eta_brush		9.57751e-007	Å ⁻²
eta_solv		6.33e-006	Å ⁻²
xsolv_core	✓	0.57	
Rg	✓	76.5148	Å
d		1	

Here for comparison, the spherical and elongated (cylindrical) micelles models have been fitted for the control conjugate (13). From the data we can observe the control conjugates fit best to an elongated micelle structure – see above for chi values. Particularly of note is the q dependence in the Guinier region (relating to morphology) best fits the elongated (hairy cylinder) model.

The molecular weight of the conjugates are not significantly different. From the molecular weight calculated for the polymer from NMR conversion shows the two polymer pBA₁₁-b-pDMA₄₁ and pDMA₅₅ are 5613 g mol⁻¹ and 5691 g mol⁻¹ respectively. The conjugates have a calculated molecular weight of 12,476 g mol⁻¹ and 12426 g mol⁻¹ for the diblock and control conjugates respectively. So for example at 1mg/mL the concentration are 8.02x10⁻⁵ M and 8.05x10⁻⁵ M respectively for the diblock and control conjugate solutions. Changing either the concentration of the conjugates or the length of the chains will also change the self-assembly of the nanotube. To keep all the variables the same other than the introduction of the hydrophobic polymer we feel this is the best control.

In the introduction: “Many of these systems have been optimised to form very large elongated assemblies, typically nanofibers, which create supramolecular gel networks.^{3,8-10}” Please consider including this recent paper precisely about hydrogels by the controlled 1D elongation of cyclic peptide assemblies: pH-Triggered self-assembly and hydrogelation of cyclic peptide nanotubes confined in water micro-droplets. *Nanoscale Horizons*, 2018, 3(4), 391–396. <http://doi.org/10.1039/C8NH00009C>. Also in the introduction: “Recent works have shown their use in a wide range of applications, in optoelectronics,¹¹ self-healable materials,¹² or bio-therapeutics.¹³⁻¹⁵” Please consider including a recent review about functional supramolecular 1D assemblies: *Chemical Communications*, 2017, 53(56), 7861–7871. <http://doi.org/10.1039/C7CC02997G>. We agree with the inclusion of the references suggested, which have now been added.

Please define in the paper (or legend of the figure) and give a suitable reference(s) for the two rate constants for FRET exchange (k₁ and k₂) included in Fig. 4d.

Explanation and suitable reference has been added:

“The exchange rates could be fit to a second order decay function relating to the monomer association/dissociation and nanotube coagulation/fragmentation.^{6,27}”

The nomenclature for the compounds is not very clear. Ideally, all the structures should be specified in the figures.

The referee makes a good point, to avoid any confusion compound numbers have been added through the text for clarity.

No indication of the beta sheets is reported. Infrared could be used to confirm amide bonds compromised in antiparallel (or parallel) β -sheet supramolecular arrangement.

We believe without the beta sheet hydrogen bonding the conjugates would not assemble into highly directional nanotubular structures observed via TEM and SANS. Also, extensive literature in this field has shown this stacking arrangements. Literature references regard the study of these self-assembling cyclic peptides using x-ray crystallography confirming this arrangement has now been added in the introduction. IR of the compounds were completed as suggested by the reviewer but as suspected, the amide bonds of the peptides are obscured by the amide bonds of the acrylamide repeating units of the polymer conjugates.

I found some minor typos including the Aida coranullene reference without Journal name (ref. 32).

This has now been corrected.

References

1. García-Iglesias, M., de Waal et al. Nanopatterned Superlattices in Self-Assembled C₂-Symmetric Oligodimethylsiloxane-Based Benzene-1,3,5-Tricarboxamides. *Chem. Eur. J.* **21**, 377-385 (2015).
2. Garzoni, M. et al. Effect of H-Bonding on Order Amplification in the Growth of a Supramolecular Polymer in Water. *J. Am. Chem. Soc.* **138**, 13985-13995 (2016).
3. Haedler, A.T. et al. Pathway Complexity in the Enantioselective Self-Assembly of Functional Carbonyl-Bridged Triarylamine Trisamides. *J. Am. Chem. Soc.* **138**, 10539-10545 (2016).
4. Hendrikse, S.I.S. et al. Controlling and tuning the dynamic nature of supramolecular polymers in aqueous solutions. *Chem. Commun.* **53**, 2279-2282 (2017).
5. Mansfield, E.D.H. et al. Systematic study of the structural parameters affecting the self-assembly of cyclic peptide-poly(ethylene glycol) conjugates. *Soft Matter* **14**, 6320-6326 (2018).
6. Rho, J.Y. et al. Probing the Dynamic Nature of Self-Assembling Cyclic Peptide-Polymer Nanotubes in Solution and in Mammalian Cells. *Adv. Funct. Mater.* **28**, 1704569 (2018).
7. Larnaudie, S.C. et al. Cyclic peptide-poly(HPMA) nanotubes as drug delivery vectors: In vitro assessment, pharmacokinetics and biodistribution. *Biomaterials* **178**, 570-582 (2018).
8. Fernandez-Lopez, S. et al. Antibacterial agents based on the cyclic D, L- α -peptide architecture. *Nature* **412**, 452-455 (2001).
9. Larnaudie, S.C. et al. Cyclic Peptide-Polymer Nanotubes as Efficient and Highly Potent Drug Delivery Systems for Organometallic Anticancer Complexes. *Biomacromolecules* **19**, 239-247 (2018).
10. Jayawarna, V. et al. Nanostructured Hydrogels for Three-Dimensional Cell Culture Through Self-Assembly of Fluorenylmethoxycarbonyl-Dipeptides. *Adv. Mater.* **18**, 611-614 (2006).
11. Adams, D.J. Dipeptide and Tripeptide Conjugates as Low-Molecular-Weight Hydrogelators. *Macromol. Biosci.* **11**, 160-173 (2011).
12. Ryan, D.M. & Nilsson, B.L. Self-assembled amino acids and dipeptides as noncovalent hydrogels for tissue engineering. *Polym. Chem.* **3**, 18-33 (2012).

13. Panda, J.J., Mishra, A., Basu, A. & Chauhan, V.S. Stimuli Responsive Self-Assembled Hydrogel of a Low Molecular Weight Free Dipeptide with Potential for Tunable Drug Delivery. *Biomacromolecules* **9**, 2244-2250 (2008).
14. Hartgerink, J.D., Beniash, E. & Stupp, S.I. Self-Assembly and Mineralization of Peptide-Amphiphile Nanofibers. *Science* **294**, 1684-1688 (2001).
15. Sato, K., Ji, W., Álvarez, Z., Palmer, L.C. & Stupp, S.I. Chiral Recognition of Lipid Bilayer Membranes by Supramolecular Assemblies of Peptide Amphiphiles. *ACS Biomaterials Science & Engineering* **5**, 2786-2792 (2019).
16. Zhou, S. *et al.* Bioactive peptide amphiphile nanofiber gels enhance burn wound healing. *Burns* **45**, 1112-1121 (2019).
17. Palmer, L.C., Newcomb, C.J., Kaltz, S.R., Spörke, E.D. & Stupp, S.I. Biomimetic Systems for Hydroxyapatite Mineralization Inspired By Bone and Enamel. *Chem. Rev.* **108**, 4754-4783 (2008).
18. Ghadiri, M.R., Granja, J.R., Milligan, R.A., McRee, D.E. & Khazanovich, N. Self-assembling organic nanotubes based on a cyclic peptide architecture. *Nature* **366**, 324-327 (1993).
19. Silk, M.R. *et al.* Parallel and antiparallel cyclic d/l peptide nanotubes. *Chem. Commun.* **53**, 6613-6616 (2017).
20. Danial, M., My-Nhi Tran, C., Young, P.G., Perrier, S. & Jolliffe, K.A. Janus cyclic peptide-polymer nanotubes. *Nat. Comm.* **4**, 2780 (2013).
21. Brendel, J.C. *et al.* Secondary Self-Assembly of Supramolecular Nanotubes into Tubisomes and Their Activity on Cells. *Angew. Chem., Int. Ed.* **57**, 16678-16682 (2018).
22. Hourani, R. *et al.* Processable Cyclic Peptide Nanotubes with Tunable Interiors. *J. Am. Chem. Soc.* **133**, 15296-15299 (2011).
23. ten Cate, M.G.J., Severin, N. & Börner, H.G. Self-Assembling Peptide-Polymer Conjugates Comprising (d-alt-l)-Cyclopeptides as Aggregator Domains. *Macromolecules* **39**, 7831-7838 (2006).
24. Couet, J., Samuel, J.D.J.S., Kopyshov, A., Santer, S. & Biesalski, M. Peptide-Polymer Hybrid Nanotubes. *Angew. Chem., Int. Ed.* **44**, 3297-3301 (2005).
25. Catrouillet, S. *et al.* Tunable Length of Cyclic Peptide-Polymer Conjugate Self-Assemblies in Water. *ACS Macro Lett.* **5**, 1119-1123 (2016).
26. Dartois, V. *et al.* Systemic antibacterial activity of novel synthetic cyclic peptides. *Antimicrob. Agents Chemother.* **49**, 3302-3310 (2005).
27. Markvoort, A.J., Eikelder, H.M.M.t., Hilbers, P.A.J. & de Greef, T.F.A. Fragmentation and Coagulation in Supramolecular (Co)polymerization Kinetics. *ACS Cent. Sci.* **2**, 232-241 (2016).

REVIEWERS' COMMENTS:

Reviewer #1 (Remarks to the Author):

The authors fully addressed all my comments as well as the ones from the other reviewers.

Reviewer #2 (Remarks to the Author):

This nice revised manuscript can be published as it is now.

Reviewer #3 (Remarks to the Author):

The authors have addressed well my comments. It is hoped that further investigations on this topic will address the construction of longer heterogeneous and crystalline nanotubes. Overall, the revision has improved the quality of the manuscript. The paper can be published.